# Mechanisms of Piperacillin/Tazobactam Nephrotoxicity: Piperacillin/Tazobactam-Induced Direct Tubular Damage in Mice

**DOI:** 10.3390/antibiotics12071121

**Published:** 2023-06-28

**Authors:** Jihyun Yang, Yoon Sook Ko, Hee Young Lee, Yina Fang, Se Won Oh, Myung-Gyu Kim, Won Yong Cho, Sang-Kyung Jo

**Affiliations:** 1Division of Nephrology, Department of Internal Medicine, Sungkyunkwan University School of Medicine, Kangbuk Samsung Hospital, Seoul 03181, Republic of Korea; 2Department of Internal Medicine, Korea University Anam Hospital, Seoul 02841, Republic of Korea

**Keywords:** nephrotoxicity, piperacillin/tazobactam, mitochondria

## Abstract

Piperacillin/tazobactam (PT) is one of the most commonly prescribed antibiotics for critically ill patients in intensive care. PT has been reported to cause direct nephrotoxicity; however, the underlying mechanisms remain unknown. We investigated the mechanisms underlying PT nephrotoxicity using a mouse model. The kidneys and sera were collected 24 h after PT injection. Serum blood urea nitrogen (BUN), creatinine, neutrophil gelatinase-associated lipocalin (NGAL), and renal pathologies, including inflammation, oxidative stress, mitochondrial damage, and apoptosis, were examined. Serum BUN, creatinine, and NGAL levels significantly increased in PT-treated mice. We observed increased IGFBP7, KIM-1, and NGAL expression in kidney tubules. Markers of oxidative stress, including 8-OHdG and superoxide dismutase, also showed a significant increase, accompanied by mitochondrial damage and apoptosis. The decrease in the acyl-coA oxidase 2 and Bcl2/Bax ratio also supports that PT induces mitochondrial injury. An in vitro study using HK-2 cells also demonstrated mitochondrial membrane potential loss, indicating that PT induces mitochondrial damage. PT appears to exert direct nephrotoxicity, which is associated with oxidative stress and mitochondrial damage in the kidney tubular cells. Given that PT alone or in combination with vancomycin is the most commonly prescribed antibiotic in patients at high risk of acute kidney injury, caution should be exercised.

## 1. Introduction

The combination of vancomycin and piperacillin/tazobactam (VPT) is a frequently prescribed antibiotic, especially in the intensive care unit (ICU), because of its broad-spectrum coverage and applicability to various healthcare-associated infections, including complicated urinary tract infections and pneumonia [1,2]. Nephrotoxicity is a well-known complication of vancomycin, with a reported incidence of 10%, and the aggregation of nanospheric vancomycin with uromodulin was recently found to be a mechanism of nephrotoxicity [3]. However, a recent meta-analysis demonstrated that the incidence of acute kidney injury (AKI) in patients treated with VPT is estimated to be 19.6–37.3% compared to that in those treated with other antibiotics combined with vancomycin or vancomycin only, suggesting the presence of synergistic nephrotoxicity in VPT treatment (VPT vs. vancomycin without PT, OR 1.25–5.67, *p* < 0.05) [4,5,6,7,8,9,10,11,12,13]. Despite this epidemiological association, only a few studies have addressed the possible mechanisms underlying piperacillin/tazobactam (PT) nephrotoxicity. Pais et al. failed to show the synergistic nephrotoxicity of VPT in a rat model [14]. Moreover, PT is associated with AKI in critically ill children with no significant association with VPT [15]. In addition, only a few studies have been conducted on AKI with biopsy-proven kidney damage in humans [16,17,18]. Both acute tubular necrosis and acute interstitial nephritis have been reported as possible mechanisms, but other reports also suggest that impaired tubular creatinine secretion is responsible for elevated creatinine levels and clinical AKI without parenchymal damage [17,18]. Therefore, it is necessary to investigate the underlying mechanisms of nephrotoxicants.

In this study, we established an animal model using a clinically relevant amount of intravascular PT injection and investigated the direct nephrotoxicity of PT monotherapy. To the best of our knowledge, this is the first study to reveal the mechanism underlying PT nephrotoxicity.

## 2. Results

### 2.1. PT-Provoked AKI 

Serum levels of blood urea nitrogen (BUN) and creatinine were significantly elevated on day 1 after PT injection (Figure 1A). Another kidney injury marker, NGAL, showed an increasing trend, but it was not statistically significant. There was no statistically significant difference between PT300 and PT500. Based on this, we believe that dose-dependent kidney damage is unlikely. Along with increased serum BUN and creatinine, immunohistochemical staining showed that the expression of IGFBP7, KIM-1, and NGAL was significantly elevated in the proximal and distal tubular cells, indicating tubular injury (Figure 1B,C).

### 2.2. PT-Provoked Oxidative Stress and Mitochondrial Damage Leading to Apoptosis in Tubular Cells

Immunohistochemical staining for 8-hydroxy-2′-deoxyguanosine (8-OHdG) and superoxide dismutase (SOD) showed significant increases in the tubular cells, indicating enhanced oxidative stress in PT-treated mice (Figure 2A,B). In contrast, catalase expression did not differ significantly between the groups.

Transmission electron microscopy demonstrated mitochondrial structural alterations (Figure 3). Compared with the vehicle-treated group, the mitochondria in PT-injected mice showed altered shape and size (Figure 3A vs. Figure 3B–D), damaged mitochondria with hyperfusion, an elongated and enlarged shape, and they contained vesicles of mitochondria with cristae loss. Autophagosomes containing dense materials, autolysosomes, and lysosomes with a relatively low density were also found close to the damaged mitochondria (Figure 3B–D). Mitochondrial damage was associated with a significant increase in the Bcl-2-associated X protein (Bax)/B-cell lymphoma 2 (Bcl2) ratio, and with an increase in tubular cell apoptosis in PT-treated mice (Figure 4A,B). A significant decrease in Acyl-CoA Oxidase 2 (ACOX2) also supports the idea that PT could provoke mitochondrial stress but not peroxisome proliferator-activated receptor-gamma coactivator-1 alpha (PGC-1α) (Figure 4C). An in vitro study using HK-2 cells also demonstrated a decrease in the red/green fluorescence of JC-1 in PT-treated cells, indicating a loss of mitochondrial membrane potential, which is a hallmark of apoptosis (Figure 4D). 

### 2.3. PT Does Not Provoke Inflammation

There was no overt interstitial inflammation in PT-injected mice. The immunohistochemical study demonstrated no abnormal increase in Ly6g+ neutrophils or F4/80+ macrophages compared to those in the vehicle group (Figure 5A,B)

## 3. Discussion

Understanding the mechanism of toxicity of antibiotics is crucial for several reasons. By elucidating the mechanisms of antibiotic toxicity, researchers can identify potential risk factors and develop strategies to minimize the occurrence of adverse reactions. This knowledge is essential for ensuring the safety of patients who require antibiotic treatment. Moreover, by investigating how certain structural features contribute to toxicity, scientists can design new antibiotics with improved safety profiles. This can aid in the development of more effective and safer drugs, leading to better patient outcomes and reduced side effects. Furthermore, it can help healthcare professionals monitor and manage patients effectively. By recognizing the underlying mechanisms of toxicity, clinicians can identify the early signs of adverse reactions and take appropriate measures to mitigate them. This understanding allows for personalized treatment plans, minimizing harm to patients and maximizing the benefits of antibiotic therapy.

In this study, we demonstrated that PT directly induces oxidative stress and mitochondrial damage, leading to tubular cell apoptosis and renal dysfunction. Our data support recent clinical observations that PT may cause nephrotoxicity.

We injected intravenous PT into mice and found that it provoked substantial tubular damage. Although overt tubular necrosis or inflammation was not found, tubular expression of injury markers, including insulin-like growth factor-binding protein 7 (IGFBP7), kidney injury molecule-1 (KIM-1), and neutrophil gelatinase-associated lipocalin (NGAL), increased significantly in proximal and distal tubular cells. Electron microscopy showed mitochondrial injury, and this was associated with increased oxidative stress and the increased number of tubular cells undergoing apoptosis with increased Bax protein expression. Direct tubular cell damage mediated by mitochondrial stress is thought to be one of the mechanisms of PT nephrotoxicity. Considering the recent revolutionary achievement of artificial intelligence (AI) in predicting the COVID-19 pandemic, AI could be employed to facilitate a better prediction of drug toxicity [19].

PT, one of the most widely prescribed antibiotics, is frequently combined with vancomycin and is generally well tolerated. However, recent meta-analyses have consistently demonstrated that the incidence of AKI in VPT-treated patients is significantly higher than that in patients treated with vancomycin monotherapy, vancomycin + meropenem, or cefepime, suggesting that PT may cause direct nephrotoxicity. Patients receiving a combination of multiple antibiotics, especially VPT, are common in ICU, and they are highly susceptible to AKI due to the severity of the disease itself and comorbidities. However, to date, only a few studies have addressed the causal relationships or underlying mechanisms. In addition, considering that a renal biopsy is rarely performed in the context of suspected acute tubular necrosis caused by an infection, the biological plausibility of PT nephrotoxicity remains unclear. Uncovering the mechanisms of antibiotic toxicity is important for ensuring drug safety, facilitating new drug design, guiding monitoring, and optimizing patient management. This knowledge enhances our understanding of antibiotics as therapeutic agents, ultimately leading to improved healthcare outcomes.

PT is a combination of drugs with piperacillin, ureidopenicillin and tazobactam, and ß-lactamase inhibitor [20,21]. The combination ratio is 8:1, and it is effective in the treatment of polymicrobial infections, including those caused by bacteria producing β-lactamases. It has broad-spectrum bactericidal activity including Gram-positive and Gram-negative bacteria and anaerobes, except Xanthomonas maltophilia. It is an effective antibiotic in treating respiratory tract, intra-abdominal, urinary tract, gynecological, and skin/soft tissue infections, and fever in neutropenic patients. The most common side effects include gastrointestinal symptoms and skin reactions. The volume of piperacillin distribution at steady state ranged from 15 to 21 L in healthy volunteers and patients, whereas that of tazobactam ranged from 18 to 34.6 L, and they are 20% plasma protein-bound. The drug distribution occurs within 30 mins after the end of infusion, and penetrates well into body tissues and fluids, with a maximum distribution occurring 1 to 2 h after infusion. The mean plasma elimination half-life is about 0.8 to 1 h, and is two-fold higher in burn patients. There is no evidence that multiple administration occurs in drug accumulation. A total of 50–60% is eliminated by renal excretion, and < 2% by biliary excretion. Therefore, patients with renal impairments require dose reduction, especially for creatinine clearance (CrCL) under 20 mL/min (<1.2 L/h). Hemodialysis removes PT, so an additional dose should be given after each dialysis session, but not in peritoneal dialysis, and there are no dosage adjustments in liver cirrhosis. It is generally administered intravenously, but intramuscular administration is also applicable for milder infections. Recently, multiple organ-support devices were applied more in ICU care, and these supports alter the pharmacokinetics of PT. Extracorporeal membrane oxygenator (ECMO) decreases the volume of distribution of the PT. Prolonged or continuous PT infusion can achieve the treatment target in critically ill patients with ECMO [22]. Extended PT infusion could achieve a greater probability of target attainment in patients who undergo continuous veno-veno hemodialysis or continuous renal-kidney replacement therapy [23]. Furthermore, Hahn et al. reported that continuous infusion of 24 g/day should be considered in ICU patients with CrCL of over 60 mL/min [24]. 

We demonstrated that PT resulted in a significant increase in BUN and creatinine levels in mice. However, there was no overt tubular necrosis in the PAS-stained tissue section, suggesting that creatinine elevation might result from the reduction of tubular creatinine secretion caused by inhibition of the organic anion transporter (OAT), as previously suggested [25]. However, we observed that serum levels of NGAL, a well-known tubular injury marker, increased in PT-treated mice. The increased expression of IGFBP7 and KIM-1 in the proximal tubules and NGAL in the distal tubules in the kidney sections further supports the occurrence of tubular injury. These data are consistent with those of a recent human study demonstrating elevated levels of urinary [TIMP2][IGFBP7] in patients who developed stage 2–3 AKI after PT or VPT administration [26,27]. 

Given that oxidative stress and mitochondrial injury are common events in drug-induced toxicity and that the kidneys are particularly rich in mitochondria, we examined oxidative stress and mitochondrial integrity in PT-treated mice. We observed that 8-OHdG, a well-known biomarker of oxidative stress, was increased in the tubular cells of PT-treated mice. In addition, the levels of SOD, an important antioxidant defense system, were also significantly increased. Despite no difference in catalase expression, these data showed that PT-induced oxidative stress in kidney tubular cells. 

The mitochondria in PT-treated mice also showed structural alterations, including enlargement, clumped cristae, and convoluted contours, which may impair cellular energetics [28,29,30,31]. Direct mitochondrial damage was further supported by our in vitro experiments, which showed a decreased red-to-green ratio of JC-1 in PT-treated HK-2 cells, indicating a loss of mitochondrial membrane potential. The significantly decreased mRNA expression of ACOX2 in PT-treated mice further supports the idea that PT induces mitochondrial injury. 

It is well known that mitochondrial depolarization unequivocally leads to apoptosis. We also observed a significant increase in TUNEL-positive apoptotic tubule cells, which was associated with decreased Bcl-2 and increased BAX protein expression. These data indicate that mitochondrial injury and subsequent apoptosis contribute to PT-induced kidney injury. Mitochondrial damage weakens renal function and impairs the ability to effectively respond to subsequent direct injuries. When mitochondrial damage occurs, it disrupts the normal energy production and metabolic processes within the renal cells. This impairment weakens the overall renal function, compromising its ability to perform essential tasks such as maintaining fluid and electrolyte balance, filtering waste products, and regulating blood pressure. Moreover, mitochondrial dysfunction also hampers the cell’s ability to respond appropriately to subsequent direct injuries. When the kidneys experience additional insults, such as ischemia (reduced blood flow) or other nephrotoxicant exposure, they rely on efficient cellular mechanisms to repair the damage and restore normal function. However, compromised mitochondria are unable to effectively execute these reparative processes. As a result, the kidneys become more susceptible to further injury, exacerbating the overall renal dysfunction.

Inflammation is also recognized as an important player in various drug-induced nephrotoxicities, including those induced by cisplatin. In addition, many beta-lactam antibiotics are well known to provoke allergic interstitial nephritis [32,33]. However, no increase in the numbers of Ly6G^+^ neutrophils or F4/80^+^ macrophages was observed in our study, suggesting that inflammation is not an important factor in PT-associated AKI. The consistent rise in renal function markers without inflammatory reactions suggests that there is a direct toxic effect due to nonimmune mechanisms. 

This study has some limitations. First, only an immediate toxicity evaluation on day 1 was conducted. How many days the toxicity lasted was not evaluated and accumulated over time. Second, this was an experiment in which the population of animals was very small. The main reason is that IACUC, an animal experiment approval institution, recommends a minimum number of animals for research. We secured the statistical significance of the results, but this study was conducted on a small number of animals. Third, we did not provide human samples and PT-related nephrotoxicity. It can be very dangerous to conduct a kidney biopsy in patients with acute kidney injury, especially in sepsis cases where antibiotics are being used, so it is necessary to fully prepare for future clinical trials in humans. In particular, it would be clinically significant if kidney biopsy could include both PT alone and VPT and others. As demonstrated in the study on vancomycin cast nephropathy, if a direct assessment of PT distribution using PT-specific antibodies and the replication of renal toxicity mechanisms observed in humans can be achieved, it could contribute to the establishment of future strategies for the prevention and mitigation of toxic effects. Finally, this study did not determine how long the renal toxicity of PT lasts. After PT injection, animals were sacrificed on day 1 to first check whether there was immediate kidney toxicity. Prolonged monitoring of PT nephrotoxicity could also be helpful to support clinical guidance.

This is the first study to show that PT is nephrotoxic and may account for the higher incidence of AKI in the VPT combination than in vancomycin monotherapy or the vancomycin–cefepime combination. Using a clinically relevant dose of PT in mice, we demonstrated that PT-induced oxidative stress and mitochondrial injury, leading to the apoptosis of tubular cells, is the key underlying mechanism of nephrotoxicity. However, functional deterioration measured by serum creatinine, which was far worse than histological damage, suggests that other mechanisms, such as decreased tubular secretion of creatinine due to the inhibition of OAT, are still possible and require further study.

## 4. Materials and Methods

### 4.1. Animal Study

Six-week-old C57BL/6 mice (Orient Bio, Inc., Seoul, Republic of Korea) were used in this study. They were housed five mice per cage, in a specific pathogen-free facility under a 12 h light/dark cycle with free access to sterile feed and distilled water. The administered dose was selected based on the clinically relevant dose for humans. Based on the usual daily dose of 12 g per 1 g in humans (150–300 mg of piperacillin/kg), 300 and 500 mg/kg were initially tested, and 300 mg/kg was used throughout the experiment because of the lack of a dose-dependent effect. Blood sampling was performed 24 h later under intra-abdominal anesthesia. Serum BUN and creatinine levels were measured using a Beckman AU® 5821 chemistry analyzer (Beckman Coulter, Brea, CA, USA) according to the manufacturer’s instructions. Serum NGAL was measured using a mouse Lipocalin-2/NGAL DuoSet enzyme-linked immunosorbent assay development kit (R&D Systems, Minneapolis, MI, USA). The kidneys were harvested. All experimental protocols were approved by the Animal Care Committee of the Korea University (IRB no.: KOREA-2018-0181). 

### 4.2. Histological Analysis

Tubular injury was semiquantitatively assessed in PAS-stained kidney tissues. To detect monocytes/macrophages or neutrophils, formalin-fixed, paraffin-embedded kidney tissues were stained with monoclonal antibodies against F4/80 (1:100; Bio-Rad Laboratories, Hercules, CA, USA) and Ly6G (1:200; eBioscience, San Diego, CA, USA). The mean number of positive cells per 8–10 high-power fields (HPFs) was compared. Tubular cell apoptosis was quantified by counting the terminal deoxynucleotidyl transferase dUTP nick-end labeling (TUNEL)-positive epithelial cells in 8–10 HPFs (200×). NGAL (1:1000; Abcam, Cambridge, UK), anti-8-OHdG antibody [N45.1] (ab48508; ABcam, Cambridge, UK), SOD (1:800; MA1-105, ThermoFisher, Berkeley, MO, USA), catalase (1:400, PA5-29183, ThermoFisher, Waltham, MA, USA), KIM-1 (1:200, MBS2006453, MyBioSource, San Diego, CA, USA), and IGFBP7 (1:1000; Abcam, Cambridge, UK)-positive cells were semiquantified using the ImageJ software (National Institutes of Health, Bethesda, MD, USA) [34]. 

### 4.3. Western Blot Analysis

Proteins were extracted from whole kidney tissues using the bicinchoninic acid method [35]. The expression of the apoptosis pathway was examined using antimouse antibodies against Bax (1:200; #2772; Cell Signaling Technology, Danvers, MA, USA) and Bcl-2 (1:200; #2762; Cell Signaling Technology, Danvers, MA, USA). The band intensities were measured using the Image Studio^TM^ Lite software (LI-COR Biosciences, Lincoln, NE, USA). Target protein levels were normalized to actin.

### 4.4. Electron Microscopy

Kidney tissue was harvested and fixed in 2% paraformaldehyde and 2.5% glutaraldehyde in 0.1 M phosphate buffer (pH 7.4) at 4 °C overnight after decapsulation. The samples were washed twice with phosphate buffer, postfixed with 1% osmium tetroxide for 2 h, rinsed with distilled water for 5 min, dehydrated stepwise with increasing concentrations of ethanol (60–100%), and embedded in an Epon mixture. The embedded samples were incubated in a 65 °C dry oven for 48 h to polymerize the resin and then trimmed. Next, 1 µm semithin sections were sliced using an ultramicrotome (Leica EM UC7; Wetzlar, Germany), collected on glass slides, stained with toluidine blue for light microscopic observation (Carl Zeiss, Oberkochen, Germany), and double stained with uranyl acetate and lead citrate. Electron microscopy analysis of morphological changes was performed using a Hitachi H-7650 electron microscope (Hitachi High-Technology Co., Tokyo, Japan).

### 4.5. Quantitative Reverse Transcription Polymerase Chain Reaction (qRT-PCR)

For detecting kidney ACOX2 and PGC1-α mRNA expression levels, total RNA was purified by TRIzol extraction (Thermo Fisher Scientific, Waltham, MA, USA) according to the manufacturer’s protocol, and complementary DNA was synthesized using standard procedures. qRT-PCR was performed in an iCycler IQ Real-Time PCR Detection System (Bio-Rad Laboratories) using iQ™ SYBR^®^ Green Supermix (Bio-Rad Laboratories) for forkhead box P3 protein and TaqMan PCR Master Mix for the others. In addition, 18S ribosomal RNA was used as the reference gene (RT2 PCR Primer Set; Applied Biosystems, Foster City, CA, USA), and fold differences relative to the vehicle group were compared.

### 4.6. In Vitro Study

To assess mitochondrial damage, 1 × 10^5^ human kidney-2 cells (HK-2) cells were cultured in 0.045 mg/mL PT for three days. JC-1 dye (lyophilized) (MitoProbe ™ JC-1 Assay Kit; M34152, Thermo Fisher Scientific, Waltham, MA, USA) was used to assess mitochondrial membrane integrity. After coculture, we added the JC-1 dye, incubated at 37 ºC for 15–30 min, and performed flow cytometric analysis (BD FACSLyric™; BD Biosciences, Franklin Lakes, NJ, USA) according to the manufacturer’s protocol. 

### 4.7. Statistical Analysis 

Data are presented as individual symbols, and the mean values are expressed as horizontal lines. A nonparametric Kruskal–Wallis test with Bonferroni’s post hoc analysis alpha 0.05 was conducted for comparisons between the two groups. ANOVA was used between the three groups, and also Bonferroni’s multiple comparison test was conducted with each post hoc analysis. We used the GraphPad Prism version 9.0 software (GraphPad Software Inc., La Jolla, CA, USA). Statistical significance was set at *p*-value < 0.05.

## 5. Conclusions

Our in vivo and in vitro studies demonstrate that PT has direct nephrotoxicity and that it is mediated via oxidative stress, mitochondrial injury, and the subsequent apoptosis of tubular epithelial cells. Mitochondrial damage in the kidneys not only directly weakens renal function but also impedes the ability of renal cells to respond adequately to subsequent injuries. Caution should be exercised when prescribing PT in combination with vancomycin for the treatment of various infections, especially in patients at a high risk of AKI.

## Figures and Tables

**Figure 1 antibiotics-12-01121-f001:**
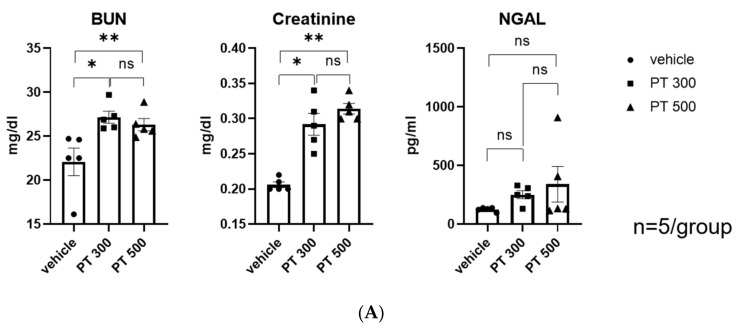
Piperacillin/tazobactam-induced renal injury. (**A**) BUN, creatinine, and NGAL levels. (**B**) Representative images of IGFBP7, NGAL, and KIM-1 immunohistochemistry-stained kidney tissue sections with quantitative analysis (100×). (**C**) Semiquantitation of IGFBP7-, NGAL-, and KIM-1-positive area. Each dot represents an individual animal, and the bar graphs show the mean values. Data are expressed as means ± standard errors of the means. The two-tailed unpaired *t*-test was used for all graphs (n = 3–5 per group). * *p*  <  0.05 compared to the vehicle group. ** *p* < 0.05 compared to the vehicle vs PT500. ns: statistically not significant. Abbreviations: PT: piperacillin/tazobactam; BUN: Serum blood urea nitrogen; NGAL: neutrophil gelatinase-associated lipocalin; IGFBP7: insulin-like growth factor-binding protein 7; KIM-1: kidney injury molecule-1.

**Figure 2 antibiotics-12-01121-f002:**
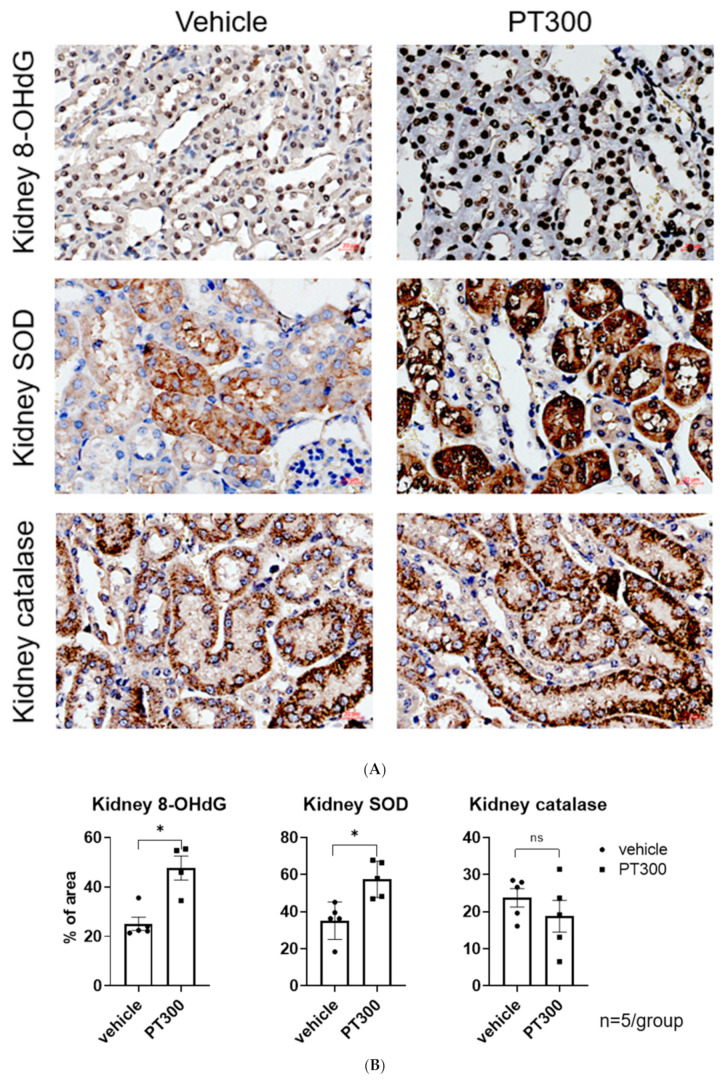
Piperacillin/tazobactam-provoked oxidative stress. (**A**) Representative images of kidney oxidative stress using 8-OHdG, SOD, and catalase. (**B**) Semiquantitation of 8-OHdG-, SOD-, and catalase-positive area per high-power field (n = 3–5 per group). * *p*  <  0.05 compared to the vehicle group. ns: statistically not significant. Abbreviations: PT: piperacillin/tazobactam; 8-OHdG: anti-8-hydroxy-2′-deoxyguanosine antibody; SOD: superoxide dismutase.

**Figure 3 antibiotics-12-01121-f003:**
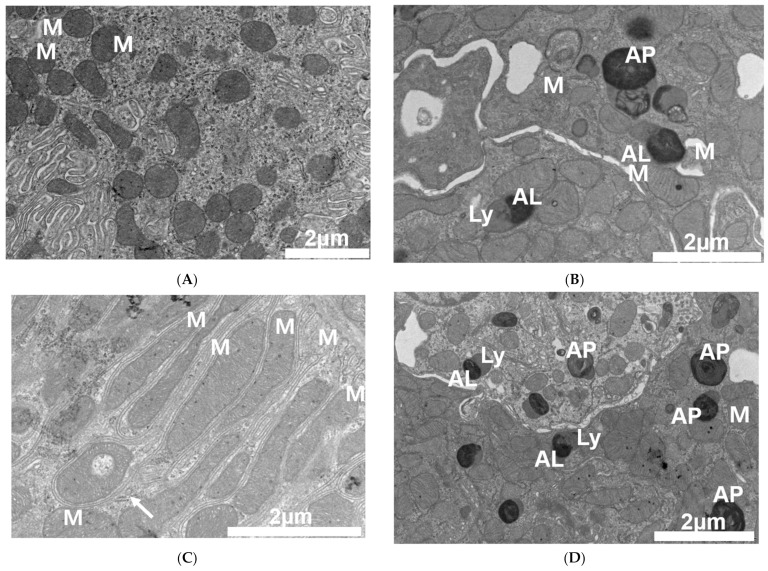
Piperacillin/tazobactam-induced renal mitochondrial injury. Electron microscopy was used to identify mitochondrial damage and the ultrastructural effects of PT nephrotoxicity. (**A**) Normal sizes and shapes of mitochondria in proximal tubular cells in vehicle group. (**B**–**D**) Mitochondria with hyperfusion and elongated and enlarged shape, vesicles inside mitochondria (arrow), and autophagosomes, autolysosome, and lysosomes with relatively low density. Abbreviations: AP: autophagosome; AL: autolysosome; L: lysosome; M: mitochondria; arrow: vesicles. Size bars: 2 μm.

**Figure 4 antibiotics-12-01121-f004:**
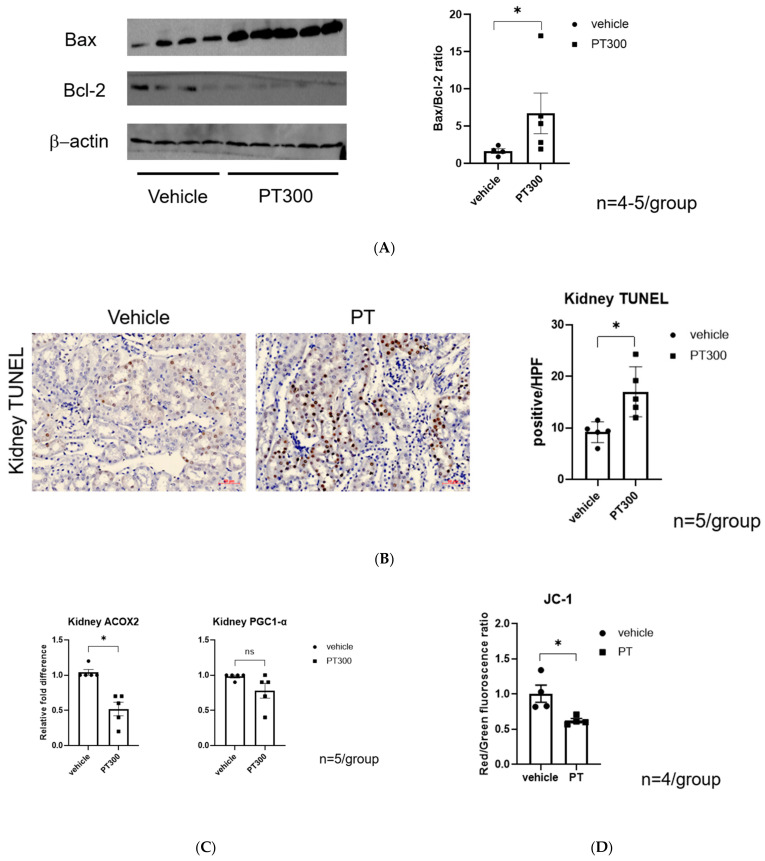
Piperacillin/tazobactam-induced renal tubular cell apoptosis. (**A**) Representative Western blot analysis images of B-cell lymphoma 2 (Bcl-2), Bcl-2-associated X protein, and β-actin. The graph shows the band intensity ratio between vehicle vs. PT. (**B**) Representative images of kidney apoptosis using TUNEL, and the number of TUNEL-positive apoptotic tubular cells (100×). The graph shows the statistical result using semiquantitation of TUNEL-positive cells per high-power field. (**C**) Real-time quantitative reverse transcription polymerase chain reaction (RT-PCR) of mitochondrial genes, including ACOX2 and PGC-1. The graph shows the fold difference compared to vehicle-treated mice. (**D**) In vitro study of JC-1 dye fluorescence ratio using HK-2 cells. * *p*  <  0.05 compared to the vehicle-treated group. ns: statistically not significant.Each dot represents an individual animal, and the bar graphs show the mean values. Data are expressed as means ± standard errors of the means. The two-tailed unpaired *t*-test was used for all graphs (n = 3–5 per group). * *p*  <  0.05 compared to the vehicle group. Abbreviation: PT: piperacillin/tazobactam; ACOX2: acyl-CoA oxidase 2; PGC1-α: peroxisome proliferator-activated receptor-gamma coactivator 1-alpha; TUNEL: terminal deoxynucleotidyl transferase dUTP nick-end labeling.

**Figure 5 antibiotics-12-01121-f005:**
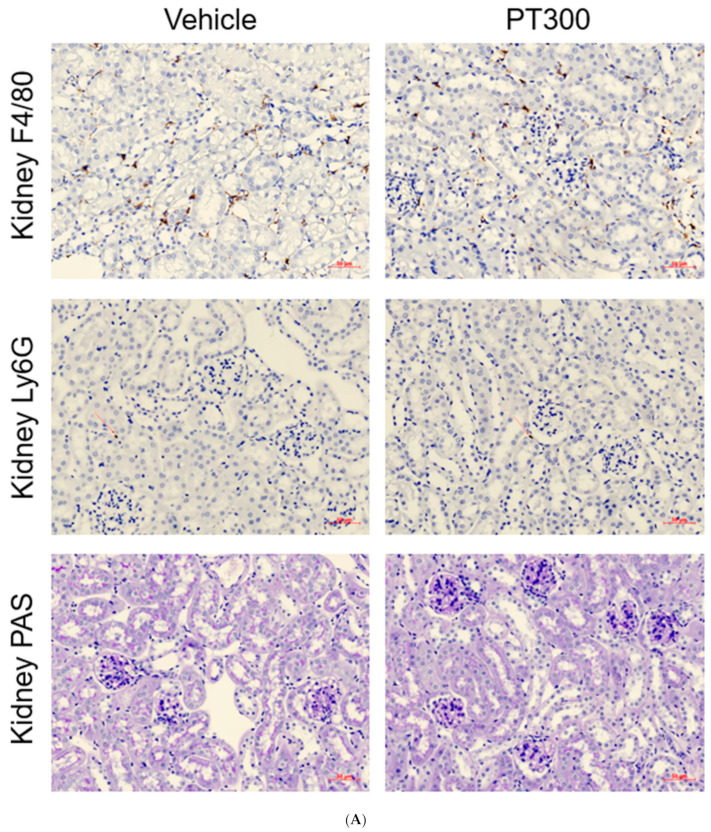
Piperacillin/tazobactam does not induce tubular necrosis or inflammation. (**A**) Representative images of F4/80, Ly6G, and PAS immunohistochemistry-stained kidney tissue sections with quantitative analysis (100×). (**B**) Semiquantitation of F4/80 and Ly6G-positive cell number per high-power field and ATN scoring using PAS histology. Each dot represents an individual animal, and the bar graphs show the mean values. Data are expressed as means ± standard errors of the means. The two-tailed unpaired *t*-test was used for all graphs (n = 3–5 per group). ns: statistically not significant. Each dot represents an individual animal, and the bar graphs show the mean values. Abbreviation: PT: piperacillin/tazobactam; PAS: periodic acid–Schiff-stained kidney tissue sections.

## Data Availability

Upon request, provision may be considered through internal evaluation.

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
