# Peer review of "Mechanisms of Piperacillin/Tazobactam Nephrotoxicity: Piperacillin/Tazobactam-Induced Direct Tubular Damage in Mice"

_antibiotics, 2023, doi:10.3390/antibiotics12071121_

Round 1
Reviewer 1 Report (New Reviewer)
Authors conducted the preclinical study by using mouse model to analyze the underlying mechanisms of nephrotoxicity induced by of Piperacillin/Tazobactam (PT) which are the most common prescribed antibiotic regimen in intensive care units. Study concluded that PT has capability to cause nephrotoxicity by directly damaging the mitochondria of the renal tubular cells which is associated with oxidative stress particularly in the presence of vancomycin. The results of the study are clinically relevant and should be considered as public health concern after further clinical trials. However, I suggest following minor changes to improve the quality of the manuscript for readers and research community. Manuscript can be accepted after incorporation of suggestions.
Comments:
Comment 1: Authors should review manuscript grammatically (Line 17-19, 25-26, 37-39). It would be better to format in vitro, in vivo terms in italics to maintain uniformity (Line 104, 125, 181, 280). Sentence should be modified for better understanding (line 280-282). It is suggested to improve the sentence structures throughout the manuscript.
Comment 2: It would be more appropriate if authors add the exact number of mice used in the study. Authors should also describe n=3-5/group. No. of animals/group should be constant instead of in range. Sample size (n=3-5/group) is relatively small for this study. It should be included in limitations section of the study and discussed it properly. .
Comment 3: The safety profile of the drug should also be discussed in discussion section.
Comment 4: Authors should add abbreviation list in the manuscript.
Comment 5: Limitation of the study should be discussed in limitation section of the manuscript.
Comment 6: Authors should add the Future perspective of the study. Manuscripts lack the implications of findings of the study. It would be more appropriate if authors add its clinical significance.
Authors should review manuscript grammatically. It is suggested to improve the sentence structures throughout the manuscript.
Author Response
Comment 1: Authors should review manuscript grammatically (Line 17-19, 25-26, 37-39). It would be better to format in vitro, in vivo terms in italics to maintain uniformity (Line 104, 125, 181, 280). Sentence should be modified for better understanding (line 280-282). It is suggested to improve the sentence structures throughout the manuscript.
Thank you for your comments. I have already received English proofreading, but I have reviewed it once again in order to get further improvement. In vitro and in vivo terms have changed the format to italic style.
Comment 2: It would be more appropriate if authors add the exact number of mice used in the study. Authors should also describe n=3-5/group. No. of animals/group should be constant instead of in range. Sample size (n=3-5/group) is relatively small for this study. It should be included in limitations section of the study and discussed it properly.
Thank you for your comments. The IACUC recommends the minimum animal number for study. We added the point that you made in the limitation section although we got the statistical significance of the result.
Comment 3: The safety profile of the drug should also be discussed in the discussion section.
Thank you for your comments. We added the safety profile ot the piperacillin/tazobactam in the discussion section as below;
Piperacillin/tazobactam is a combination of drugs with piperacillin, ureidopenicillin and tazobactam, ß-lactamase inhibitor. The combination ratio is 8:1, it is effective in the treatment of polymicrobial infections including those caused by bacteria producing β-lactamases. It has broad spectrum bactericidal activity including Gram-positive, Gram-negative bacteria, and anaerobes except Xanthomonas maltophilia. It is an effective antibiotic in treating respiratory tract, intra-abdominal, urinary tract, gynecological, and skin/soft tissue infections, as well as fever in neutropenic patients. The most common side effects include gastrointestinal symptoms and skin reactions. The volume of piperacillin distribution at steady state ranged from 15 to 21L in healthy volunteers and patients, tazobactam was 18 to 34.6L, they are 20% plasma protein bound. The drug distribution occurs within 30 minutes after the end of infusion, well penetrating into body tissues and fluids the maximum distribution occurred 1 to 2 hours after infusion. The mean plasma elimination half-life is about 0.8 to 1 hour, 2 fold higher in burn patients. There is no evidence that multiple administration occurs drug accumulation. It is eliminated 50-60% by renal excretion, < 2% biliary excretion. Therefore, patients with renal impairments require dose reduction especially creatinine clearance (CrCL) under 20 ml/min (<1.2L/hr). Hemodialysis removes PT so an additional dose should be given after each dialysis session, but not in peritoneal dialysis, no dosage adjustments in liver cirrhosis. It is generally administered intravenously, but intramuscular administration is also applicable for milder infection. Recently, multiple organ-support devices apply more and more in ICU care, and those supports alter the pharmacokinetics of PT. Extracorporeal membrane oxygenator (ECMO) decreased the volume of distribution of the PT. Prolonged or continuous PT infusion can achieve the treatment target in critically ill patients with ECMO. Extended PT infusion could achieved greater probability of target attainment in patients continuous veno-veno hemodialysis or continuous renal-kidney replacement therapy. Hahn et al said that continuous infusion 24g/day should be considered in ICU pateints with CrCL over 60 ml/min.
Comment 4: Authors should add abbreviation list in the manuscript.
Thank you for your comments. We separately added the abbreviation list at the end of the manuscript;
AKI; acute kidney injury,
PT; Piperacillin/tazobactam
BUN; blood urea nitrogen
NGAL; neutrophil gelatinase-associated lipocalin
VPT; vancomycin and piperacillin/tazobactam
OR; odds-ratio
IGFBP7; insulin-like growth factor-binding protein
KIM-1; kidney injury molecule-1
8-OHdG; 8-hydroxy-2'-deoxyguanosine
SOD; superoxide dismutase
ACOX2; Acyl-CoA Oxidase 2
HK-2 cell; human kidney-2 cell
Bax; Bcl-2-associated X protein
Bcl2; B-cell lymphoma 2
PGC-1-alpha; proliferator-activated receptor-gamma coactivator-1 alpha
TUNEL; terminal deoxynucleotidyl transferase dUTP nick-end labeling
PAS, periodic acid–Schiff-stained kidney tissue sections
OAT; the organic anion transporter
Comment 5: Limitation of the study should be discussed in limitation section of the manuscript.
Thank you for your comments. We added the limitation as below;
This study has the following limitations. First, only the immediate toxicity evaluation that occurred on day 1 was conducted. How many days the toxicity lasted was not evaluated and accumulated over time. Secondly, it was an experiment in which the population of animals was very small. The main reason is that IACUC, an animal experiment approval institution recommends a minimum number of animals for research. We secured the statistical significance of the results, but this study was conducted in very few animals. Third, we did not provide human samples and PT-related nephrotoxicity. It can be very dangerous to conduct a kidney biopsy in patients with acute kidney injury, especially in sepsis cases where antibiotics are being used, so it is necessary to fully prepare for future clinical trials in humans. In particular, it would be clinically significant if kidney biopsy could include both PT alone and VPT and others. As demonstrated in the study on vancomycin toxicity, if direct assessment of PT distribution using PT-specific antibodies and replication of renal toxicity mechanisms observed in humans can be achieved, it could contribute to the establishment of future strategies for the prevention and mitigation of toxic effects.
Comment 6: Authors should add the Future perspective of the study. Manuscripts lack the implications of findings of the study. It would be more appropriate if authors add its clinical significance.
Thank you for your comments. We added the implication and clinical significance of this study as above.
Reviewer 2 Report (New Reviewer)
sample size too small to perform statistical analysis. Case series should be applied with six participants, even animals.
Author Response
Comments and Suggestions for Authors
sample size too small to perform statistical analysis. Case series should be applied with six participants, even animals.
Thank you for your comments. We clearly mentioned at the discussion limitation section about the small number of the animal.
J Charan, N.D Kantharia recommended using E valuae between 10 and 20 for setting the statistical power for animal experiments. (reference : “How to calculate sample size in animal studies?” (Journal of Pharmacology and Pharmacotherapeutics | October-December 2013 | Vol 4 | Issue 4). They pointed out that five per group is acceptable limit. Arifin and Zahiruddin also said, between five and seven per group would be required (reference: “Sample Size Calculation in Animal Studies Using Resource Equation Approach”, Malays J Med Sci. Sep–Oct 2017; 24(5): 101–105)
According to engage animal rights, the animal IRB practice, IACUC recommend us minimizing experiment animals to as much as 5 per group is acceptable. As we described in the first part of the result section, the total group was set 3 (vehicle, PT 300, PT 500), renal function finding was not significantly different between PT 300 vs PT 500, so we focused on vehicle and PT 300. We added more vehicles to make sure that n=5 in each group.
Reviewer 3 Report (New Reviewer)
The work carried out by Jihyun Yang et al. consists of a preclinical evaluation of nephrotoxicity and the mechanisms associated with it caused by the drug combination piperacillin/tazobactam. The findings obtained are very interesting and demonstrate a clear nephrotoxicity related mainly to processes of oxidative stress and mitochondrial damage. Overall, it is well written and detailed, but from my point of view it requires a number of corrections and improvements before it is finally published:
- Introduction, lines 51-61: I think that all this information should not appear in the introduction of the work. It is basically a summary of the experiment carried out and the results obtained, not information corresponding to the introduction.
- Figure 1A: the statistical significances obtained between the "Vehicle" group and the "PT 500" group are not shown.
- Figure 1C and the rest of the figures: I imagine that the "PT" group is the "PT 300" group. I recommend always including the full name so as not to mislead with the "PT 500" group.
- Some figures should be reviewed (Figure 4 and Figure 5), as there are remains of the graphic edition (a red underline under Bax, some frames around "n=3-5/group", for example).
- Figure 5 caption: there is a repeated paragraph: "Data are expressed as means ± standard errors of the means. The two-tailed unpaired t-test was used for all graphs (n = 3-5 per group). *P < 0.05 compared to the vehicle group."
- Section 4.1. The experimental design is not well understood (it is necessary to detail which experimental groups are included, including the vehicle, explain the doses used and justify their choice). It is also not very well understood why the PT 500 dose is not used later and why more studies are not carried out with these animals.
- Section 4.1. Why is the sacrifice carried out at 24 h and it is not decided to extend the study longer? I think you should justify this.
- Lines 236-137: the method should be described or include a citation to a work that describes it.
- Statistical analysis: has the normality of the data been previously evaluated? if the data do not adjust to normality, it is not possible to apply the t-test (and in any case, when n=3 it is preferable to apply non-parametric tests, since normality cannot be evaluated). In addition, it is not indicated which tests have been applied when the three study groups are included (the t test is not possible, since it only allows comparing two groups).
The wording of the paper should be reviewed, as there is a small error in the writing (for example, "damagedd" appears on line 95 and "//" appears on line 210).
Author Response
Comments and Suggestions for Authors
The work carried out by Jihyun Yang et al. consists of a preclinical evaluation of nephrotoxicity and the mechanisms associated with it caused by the drug combination piperacillin/tazobactam. The findings obtained are very interesting and demonstrate a clear nephrotoxicity related mainly to processes of oxidative stress and mitochondrial damage. Overall, it is well written and detailed, but from my point of view it requires a number of corrections and improvements before it is finally published:
- Introduction, lines 51-61: I think that all this information should not appear in the introduction of the work. It is basically a summary of the experiment carried out and the results obtained, not information corresponding to the introduction.
Thank you for your comments: we have rearranged this paragraph into the discussion section.
- Figure 1A: the statistical significances obtained between the "Vehicle" group and the "PT 500" group are not shown.
Thank you for your comments: added each p-values.
- Figure 1C and the rest of the figures: I imagine that the "PT" group is the "PT 300" group. I recommend always including the full name so as not to mislead with the "PT 500" group.
Thank you for your comments: changed all the PT PT 300. The in vitro study JC-1 Assay, 0.045 mg/mL PT for three days treatment was only done so it remained the same.
- Some figures should be reviewed (Figure 4 and Figure 5), as there are remains of the graphic edition (a red underline under Bax, some frames around "n=3-5/group", for example).
Thank you for your comments: we made more clear edition, also the file format is “ppt” so it Is modifiable.
- Figure 5 caption: there is a repeated paragraph: "Data are expressed as means ± standard errors of the means. The two-tailed unpaired t-test was used for all graphs (n = 3-5 per group). *P < 0.05 compared to the vehicle group."
Thank you for your comment: modfied legends.
- Section 4.1. The experimental design is not well understood (it is necessary to detail which experimental groups are included, including the vehicle, explain the doses used and justify their choice). It is also not very well understood why the PT 500 dose is not used later and why more studies are not carried out with these animals.
Please refer to the results of renal function tests, which are the most important and primary comparative evaluation metrics obtained through intravenous injections of the vehicle, PT300, and PT500. As shown in Figure 1, there was no statistically significant difference between PT300 and PT500. Based on this, we believe that dose-dependent kidney damage is unlikely. Despite administering higher doses, we did not observe additional deterioration in renal function. Therefore, we conducted our study using low doses of PT that are closer to the clinically relevant levels, which we consider as meaningful and representative of actual usage. As stated in the submission guidelines of this journal - 'When conducting research on the mechanism and toxic effects of a drug, use doses at a clinical level. Excessive and excessively high doses for evaluating toxic effects are not meaningful' - we presented only the data for PT300 to comply with this recommendation. However, if excluding the data for PT500 might cause confusion and deviate from the core focus of this journal, we can consider excluding the data for PT500.
- Section 4.1. Why is the sacrifice carried out at 24 h and it is not decided to extend the study longer? I think you should justify this.
Thank you for your comments. It is known that when mitochondrial damage occurs, it lasts for several days. However, this study did not determine how long the renal toxicity of PT lasts. After PT injection, animals were sacrificed on day 1 to first check whether there was immediate kidney toxicity. This content is described in the limitations.
- Lines 236-137: the method should be described or include a citation to a work that describes it.
Thank you for your comments. We added the reference at the BCA methods (Anal Biochem
. 1992 Aug 1;204(2):332-4)
- Statistical analysis: has the normality of the data been previously evaluated? if the data do not adjust to normality, it is not possible to apply the t-test (and in any case, when n=3 it is preferable to apply non-parametric tests, since normality cannot be evaluated). In addition, it is not indicated which tests have been applied when the three study groups are included (the t test is not possible, since it only allows comparing two groups).
Thank you for your comments. In this study with small sample size, most data showed skwed distribution, we used non-parametric Wilcoxon test between two groups. ANOVA was used between the three groups, and Tukey's multiple comparison test was conducted with each post-hoc analysis.
Round 2
Reviewer 2 Report (New Reviewer)
line 70: have you tested the beta error?
line 378: ANOVA cannot be used for not Gaussian distribution.
The sample is not large enough to perform statistical analysis. Case series should be considered as study design. If you want to perform statistical analysis, the Bayesian assumption could help you, but for the frequentist methods these numbers are not enough.
Author Response
line 70: have you tested the beta error?
Thank you for your comments. We used GraphPad prism version 9.0, standard set alpha to 0.05, and beta to 0.20.
line 378: ANOVA cannot be used for not Gaussian distribution.
Thank you for your comments. Using the Kruskal-Wallis test, still the p-value remains 0.006, with Bonferroni’s post hoc analysis alpha 0.05 applied. With the assistance of a statistical expert, a statistical analysis was re-conducted using SPSS instead of GraphPad Prism, and the results obtained are as follows. It was observed that, except for NGAL, the trends were consistent. (analysis result is attached in graphic form in word file)
The sample is not large enough to perform statistical analysis. Case series should be considered as study design. If you want to perform statistical analysis, the Bayesian assumption could help you, but for the frequentist methods these numbers are not enough.
Thank you for your comments.
In this study, a non-parametric statistical analysis was performed, and it yielded similar statistically significant results except NGAL : BUN, sham vs PT300, p-value 0.028; sham vs PT500, p-value 0.002; PT300 vs PT500, p-value 0.36, Cr, sham vs PT300, p-value 0.023; sham vs PT500, p-value 0.002, PT 300vs PT500, p-value 0.43, NGAL, sham vs PT300 p-value 0.9; sham vs PT500 p-value 0.4, PT 300 vs PT 500, p-value 0.9. The analysis was re-examined with the assistance of the statistics professor at our institution. It should be noted that the limited number of animals in this experiment, comparable to a case series, could be a limitation as you recommended.

Reviewer 3 Report (New Reviewer)
In general, all my suggestions and doubts have been resolved. However, there is one aspect that should still be reviewed:
After redoing the statistical analysis (after evaluating normality) the authors indicate that a non-parametric method was used for the comparison, which seems correct to me. However, when comparing the three groups, they still use a parametric ANOVA-Tukey test. I think that a non-parametric test should be applied to compare the three groups (taking their distribution into account), such as the Kruskal-Wallis test (with Bonferroni's post-hoc correction, for example). Otherwise, the significances assigned to each result may not be correct.
Author Response
In general, all my suggestions and doubts have been resolved. However, there is one aspect that should still be reviewed:
After redoing the statistical analysis (after evaluating normality) the authors indicate that a non-parametric method was used for the comparison, which seems correct to me. However, when comparing the three groups, they still use a parametric ANOVA-Tukey test. I think that a non-parametric test should be applied to compare the three groups (taking their distribution into account), such as the Kruskal-Wallis test (with Bonferroni's post-hoc correction, for example). Otherwise, the significances assigned to each result may not be correct.
Thank you for your comments. Using the Kruskal-Wallis test, still the p-value remains 0.006, with Bonferroni’s post hoc analysis alpha 0.05 applied. With the assistance of a statistical expert, a statistical analysis was re-conducted using SPSS instead of GraphPad Prism, and the results obtained are as follows. It was observed that, except for NGAL, the trends were consistent. In this study, a non-parametric statistical analysis was performed, and it yielded similar statistically significant results except NGAL : BUN, sham vs PT300, p-value 0.028; sham vs PT500, p-value 0.002; PT300 vs PT500, p-value 0.36, Cr, sham vs PT300, p-value 0.023; sham vs PT500, p-value 0.002, PT 300vs PT500, p-value 0.43, NGAL, sham vs PT300 p-value 0.9; sham vs PT500 p-value 0.4, PT 300 vs PT 500, p-value 0.9.
Round 3
Reviewer 2 Report (New Reviewer)
limits have been included and statistical methods have been corrected. However, i would presented it as a case series.
Author Response
Reviewer’s comment: limits have been included and statistical methods have been corrected. However, i would presented it as a case series.
Response: Thank you for your comments. I am writing to you today to respectfully request that you reconsider your concerns about this study. This research focuses on studying the toxicological mechanisms of a drug using animal experiments. It does not report on specific cases of adverse reactions. We have clearly acknowledged the limitations of this study, such as the small number of animal subjects and the need for further research.
I believe that the unique nature of this study warrants a little bit more positive review. The research is intuitively-designed and provides valuable new insights into the toxicological mechanisms of piperacillin/tazobactam. The limitations of the study are acknowledged, and they do not detract from the overall findings.
I would be grateful if you would consider these points and reconsider your concerns about this study. Thank you for your time and consideration.
This manuscript is a resubmission of an earlier submission. The following is a list of the peer review reports and author responses from that submission.
Round 1
Reviewer 1 Report
Journal: Antibiotics (ISSN 2079-6382)
Manuscript ID: antibiotics-2007362
Title: Mechanisms of Piperacillin/Tazobactam Nephrotoxicity: Piperacillin/Tazobactam induced direct tubular damage
The authors investigated the PT nephrotoxicity mechanism. Methods: We injected PT into 6-week-old C57L/B6 mice via the tail vein, and the kidneys and sera were collected 24 h later. We injected 0.9% normal saline into the sham group mice. Serum blood urea nitrogen (BUN), creatinine, neutrophil gelatinase-associated lipocalin (NGAL), renal in- flammation, apoptosis, cell death (specifically the mitochondria-related pathway), and the micro-scopic and immunohistochemical renal pathology were evaluated. Please highlight your contributions in introduction. Discuss the novelty and motivation in the last paragraph in the introduction.
Quality of Figure1 should be improved.
The caption of Figure 1 MUST be updated. What is BUN, NGAL, ….etc?
What are main features in Figure 1 B?
Quality of Figure3 should be improved.
The caption of Figure 3 MUST be updated. What is BAX, Bcl-2, ….etc?
What are main features in Figure 1 B?
Add more figures to discuss your results.
The introduction should be supported by recent publications from MDPI such as “Artificial intelligence for forecasting the prevalence of COVID-19 pandemic: an overview”.
Conclusion: What are the advantages and disadvantages of this study.
The inspiration of your work must further be highlighted.
Future work must be included.
Looking and wishes for the revised version.
Reviewer 2 Report
The authors of this study determined the Mechanisms of Piperacillin/Tazobactam Nephrotoxicity: Piperacillin/Tazobactam induced direct tubular damage. This is an interesting study and the authors have collected a dataset of results. Although, I have major concerns regarding the manuscript as mentioned below.
1. kidney injury marker-1 (KIM-1) seems to have several advantages over other markers of kidney injury. It was shown to increase in urine shortly after injury, before renal tubular damage could be observed in histological examination. Huang et al., confirmed that KIM-1 was increased within 24 h after kidney injury. Regarding the kidney injury KIM-1 is a promising marker of renal tubular injury/acute kidney injury, author need to shows the response of KIM-1 with the treatment of Piperacillin/Tazobactam.
Huang Y, Don-Wauchope AC. The clinical utility of kidney injury molecule 1 in the prediction, diagnosis and prognosis of acute kidney injury: a systematic review. Inflamm Allergy Drug Targets. 2011 Aug;10(4):260-71. doi: 10.2174/187152811796117735.
Han WK, Bailly V, Abichandani R, Thadhani R, Bonventre JV. Kidney Injury Molecule-1 (KIM-1): a novel biomarker for human renal proximal tubule injury. Kidney Int. 2002 Jul;62(1):237-44. doi: 10.1046/j.1523-1755.2002.00433.x.
2. Author also needs to show Piperacillin/Tazobactam effect on oxidative stress markers (LPO, GSH, Cat, SOD)
3. In abstract and discussion author is showing that treatment of Piperacillin/Tazobactam caused a significant decrease in the expression of PGC1-α but in results there is no significant decrease in the expression of PGC1-α.
4. Author needs to add more markers to prove Piperacillin/Tazobactam caused tubular death/nephrotoxicity through mitochondrial injury.
5. However, the available research information seems to be insufficient for being reviewed. Taken together all these issues, In the current form, the manuscript doses not fit for publication in the Journal.
Reviewer 3 Report
This important manuscript investigated biochemical and microscopic effects of use of piperacillin/vancomycin combination on renal tissue, thus trying to more precisely define nephrotoxicity potential and mechanisms.
This study is characterized by precise methodology and complex analysis which further enlarges its importance.
The manuscript itself is very well written with almost no obvious major flaws.
However, some important points need to be addressed.
It is not very clear what was the reason for selecting IGFBP7 and NGAL, since several other biomarkers of renal damage have been analyzed so far in the literature.
Also, the piperacillin vs tazobactam analysis was not performed, but the combination analysis only. It would be great if authors could explain the reasons for doing so. Besides, vancomycin is mentioned in the Introduction, as well as in the Discussion and Results sections, but it was not involved in the analysis. This issue needs to be further clarified.
One additional minor point with a need to be addressed is the fact that IGFBP7 is not mentioned as an investigated biomarker in the Abstract's Methods section, while it is present in the rest of the Abstract.
Round 2
Reviewer 1 Report
Accept in present form.
Reviewer 2 Report
According to me this data is not sufficient to prove PT induced nephrotoxicity via mitochondrial damage, needs more experiments as i mentioned in my previous comments.
However, the available research information seems to be insufficient for being reviewed. Taken together all these issues, In the current form, the manuscript doses not fit for publication in the Journal.